# Mechanical and Metallurgical Properties of CO₂ Laser Beam INCONEL 625 Welded Joints

**Harinadh Vemanaboina** [1],*, **Edison Gundabattini** [2], **Suresh Akella** [3], **A. C. Uma Maheshwer Rao** [3], **Ramesh Kumar Buddu** [4], **Paolo Ferro** [5],* and **Filippo Berto** [6]

1    Department of Mechanical Engineering, Sri Venkateswara College of Engineering and Technology (Autonomous), Chittoor 517127, India
2    Department of Thermal and Energy Engineering, School of Mechanical Engineering, Vellore Institute of Technology (VIT), Vellore 632014, India; edison.g@vit.ac.in
3    Department of Mechanical Engineering, Sreyas Institute of Engineering & Technology, Hyderabad 500068, India; s4akella@sreyas.ac.in (S.A.); umamaheswar.ac@sreyas.ac.in (A.C.U.M.R.)
4    Institute for Plasma Research, Gandhinagar 382428, India; buddu@ipr.res.in
5    Department of Engineering and Management, University of Padua, Stradella San Nicola, 36100 Vicenza, Italy
6    Department of Engineering Design and Materials, Norwegian University of Science and Technology, 7491 Trondheim, Norway; filippo.berto@ntnu.no
*    Correspondence: harinadhv@svcet.in (H.V.); paolo.ferro@unipd.it (P.F.)

**Abstract:** In the frame of the circular economy, welding of Ni-based superalloys has gained increasing importance when applied, for instance, to repairing highly expensive components widely used in strategical sectors, such as the defense and aerospace industries. However, correct process parameters avoiding metallurgical defects and premature failures need to be known. To reach this goal, Inconel 625 butt-welded joints were produced by CO₂ laser beam welding and different combinations of process parameters. The experimental investigation was carried out with three parameters in two levels with an L₄ orthogonal array. Laser power, welding speed, and shielding gas flow rate were varied, and the results were reported in terms of mechanical properties, such as microhardness, tensile strength, distortion, residual stress, and weld bead geometry, and metallurgy. At a lower welding speed of 1 m/min, the full penetration was observed for 3.0 kW and 3.3 kW laser powers. However, sound welds (porosity-free) were produced with a laser power of 3.3 kW. Overall, the obtained full-penetration specimens showed a tensile strength comparable with that of the parent material with residual stresses and distortions increasing with the increase in heat input.

**Keywords:** laser beam welding; radiography; tensile strength; residual stress; microstructure

## 1. Introduction

Welding is a highly complex process of permanently joining metals that involves heat source movement, mass exchange, and phase and microstructure transformations that in turn affect the joint's mechanical properties [1]. In response to the growing demand for high-performance materials by the chemical, nuclear, fuel, aerospace, marine, and petroleum-based industries that are extremely resistant to a corrosive atmosphere, high-strength nickel-based alloys are mainly used. They combine excellent corrosion resistance with high strength at high temperatures. Nickel-based materials are austenitic superalloys containing about 50% of Ni and different percentages of Fe, Cr, Mo, and Ti (present as trace elements). Nickel offers good thermal and corrosive resistance at elevated temperatures. The grain size at the fusion zone depends on the alloy's chemical composition, as well as welding parameters such as interaction time, welding speed, and current. The fusion welding process promotes a thermo-mechanical variation of the parent metal properties; the high heat input and uncontrolled cooling rates negatively affect the microstructure of the fusion zone (FZ) and heat-affected zone (HAZ), inducing precipitation of undesirable phases, grain coarsening (in HAZ) and tensile residual stresses, among others. Moreover,

these mentioned changes in the welded joint also have a negative impact on components' performances when working at elevated temperatures.

The gas tungsten arc welding (GTAW) process is widely employed to join high-strength alloys with suitable filler materials with both constant and pulsed current processes [2–4]. As per the earlier research, the microstructures were observed to have satisfactory solid solubility with parent and filler metals but a wide fusion zone. Hot cracking is one of the major problems observed during single-pass welding, resulting in unacceptable permanent damage to the structure. As an alternative, this was overcome by employing a multipass welding procedure with the foresight to maintain appropriate inter-pass temperature [5–8]. However, the mechanical and metallurgical properties can be improved by employing advanced welding techniques such as electron beam welding (EBW) and laser beam welding (LBW). Their high-power density allows for reducing the heat input and therefore the FZ and HAZ dimensions together with a higher welding speed. This results in improved metallurgical and mechanical properties and productivity. Moreover, these processes can be easily automated compared to conventional welding techniques.

The full-penetration welding is carried out in a key-hole regime, which results in quality welds with a narrow heat-affected zone (HAZ), as mentioned above. Excess penetrations are eliminated, and no further machining process for finishing in LBW welds compared to the arc welding process is needed. Microstructural changes are limited because of the low heat input with faster cooling rates [9–15]. In references [16,17], authors carried out a comparative study using both arc and electron beam welding process applied to 316L stainless steels; they found that metastable phases formed during solidification of fusion zone in EBW joints compared to that obtained with the GTAW process. Roshith et al. [18] focused on the joining of SMO 254 stainless steel by means of autogenous $CO_2$ laser beam welding (LBW) and pulsed current gas tungsten arc welding (PCGTAW). They found that for a 5 mm thick plate, the LBW heat input was only 120 J/mm, compared to a value of 3032 J/mm required by the PCGTAW process. Surprisingly, tensile residual stress was observed in the weld region and heat-affected zone of PCGTAW joints, while pure compressive residual stresses were induced in the case of laser-welded joints due to lower heat input value. Full penetration was observed at 120 and 180 J/mm for Inconel 825 and Inconel 625 of 5 mm-thick plates, respectively, with faster cooling rates and reduced grain growth at the fusion zone, compared to that induced by the GTAW process [9,18,19]. Finally, it is worth mentioning that process parameter optimization can also be achieved by numerical simulation [20–29] that, however, cannot disregard experimental tests to be validated.

In recent years, Ni-based alloys are growing in their strategical importance given their use in manufacturing and machining industry applications where mechanical and corrosion resistance at high temperatures are needed. However, the effect of welding process parameters on the weldability of Inconel 625 and its structural properties are not yet completely well studied in the literature, in particular when considering the recognized benefits of high-power density welding techniques. This work is aimed at contributing to fill this gap. $L_4$ orthogonal array was used for the $CO_2$ laser beam welding trials of Inconel 625 plates. The microstructural changes and thereby the corresponding mechanical properties of the final welds are analyzed in detail.

## 2. Materials and Methods

### 2.1. Parent Material and Edge Preparation

In this work, 5 mm-thick Inconel 625 plates were butt-welded. Each plate was 110 mm long and 80 mm wide. The chemical composition of the parent material is summarized in Table 1. Care was taken to prepare the samples in order to avoid any type of defect due to contamination of the material. Sample edges were therefore cleaned by a wire electrical discharge machine (WEDM). In particular, the machining was carried out on the lateral and top surfaces to remove oxide layers. Before welding, the perfect positioning of the plates, having a 90° square groove without any gap in the welding line, was accurately checked.

**Table 1.** Nominal composition of Inconel 625 (wt.%).

| Ni | C | Mn | S | Cu | Si | Cr | P | Others |
|---|---|---|---|---|---|---|---|---|
| Bal. | 0.1 | 0.5 | 0.015 | 0.5 | 0.5 | 20–23 | 0.015 | Fe 5, Al 0.40, Mo 8-10, Ti 0.1 |

### 2.2. Melt Run Trials and Butt-Welding

Melt run trials were carried out on 5 mm-thick Inconel plates using a $CO_2$ laser source in order to analyze the effects of welding parameters on geometrical characteristics and microstructure of the bead. Particular attention was paid to those parameters assuring full penetration in the subsequent butt-welding operations. The previous experiment using a laser power of 2.5 kW did not show good results (Figure 1a), and so it was decided to start with a minimum power of 3 kW, as shown in Figure 1b. $L_4$ orthogonal array approach was selected for the experimentation. Laser power, welding speed, and shielding gas flow rate were chosen as critical process parameters for the present work. Laser power and welding speed are the main parameters regulating the heat input during the welding operation. On the other hand, weldments responses are affected above all by the heat input. Four combinations of the three selected process parameters were chosen according to Table 2 (design of experimentation (DOE)). Argon was employed as shielding gas (purity equal to 99.9%) during welding. During welding operations, plates were clamped in jaws in order to assure the absence of gaps, as shown in Figure 2a. The $CO_2$ laser beam welding setup is shown in Figure 2b.

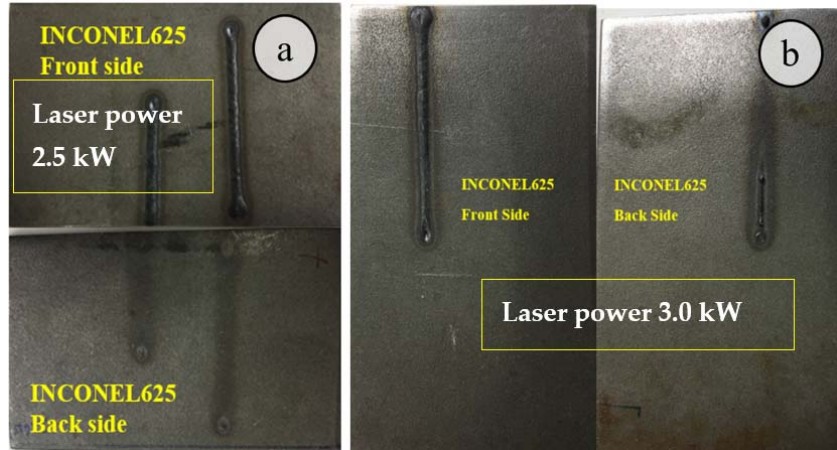

**Figure 1.** Images illustrating Inconel 625 melt run trials: (**a**) 2.5 kW laser power, showing poor penetration, (**b**) 3 kW laser power, inducing good penetration.

**Table 2.** $CO_2$ laser beam welding experimental process parameters.

| Trial No. | Laser Power [kW] | Welding Speed [m/min] | Shield Gas Flow Rate [lpm] | Heat Input [J/mm] |
|---|---|---|---|---|
| | A | B | C | HI |
| 1. | 3.0 | 1.0 | 10 | 180 |
| 2. | 3.0 | 1.5 | 15 | 120 |
| 3. | 3.3 | 1.0 | 15 | 198 |
| 4. | 3.3 | 1.5 | 10 | 132 |

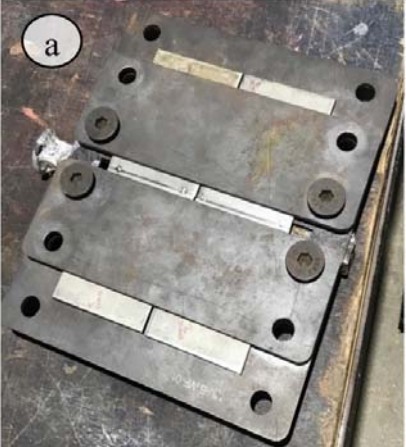 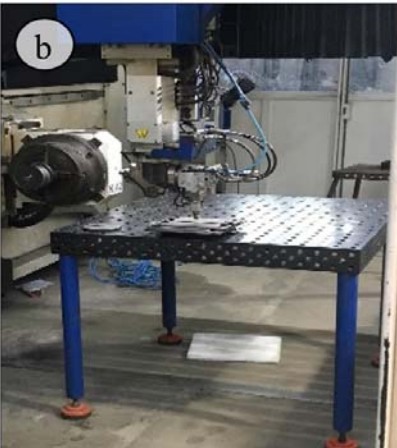

**Figure 2.** (**a**) Clamping conditions; (**b**) $CO_2$ laser beam welding machine.

The heat input (*HI*, J/mm), i.e., the ratio between the laser power and the welding speed, is calculated for each experimental trial by using Equation (1)

$$HI = \frac{60W}{V} \qquad (1)$$

where *W* is laser power (kW), and *V* is the welding speed (m/min). In general, it is well known that the lower the heat input, the better the welded joint quality in terms of bead microstructure and mechanical properties. The heat input values resulting from the combinations of the chosen values of process parameters are collected in Table 2.

### 2.3. Defects and Microstructure Investigations

Weldments were carefully analyzed using the X-ray radiographic technique to check the absence of defects such as macroporosity and lack of penetration. After non-destructive analysis, the as-welded samples were cut to produce samples for metallurgical and mechanical tests. Optical and scanning electron microscopy (with EDS) were used to carry out metallurgical investigations on the weld bead cross-section. Samples were prepared following the standard metallurgical procedure with polishing using different emery sheets and final polishing using a double-disc polisher with $Al_2O_3$ slurry. The weld zone was investigated by XRD analysis, as well. In particular, the X-ray diffraction technique was operated at 30 mA and 40 kV with a 2θ angle between 20° and 90°.

### 2.4. Welding Distortions, Residual Stresses, and Mechanical Tests

Any unwanted geometrical change or departure from specifications in a fabricated structure or component, as a consequence of welding, is called welding distortion. Temperature gradients induced by welding cause dimensional changes in the weldments that adversely affect the structure assembly. In particular, welding distortions are due to the non-homogeneous plastic deformation occurring during welding because of the constrained alloy thermal expansion and contraction. Before welding, the plates were clamped and the constraints were removed after process completion, as shown in Figure 2a. The weld distortion was measured using the facility schematized in Figure 3 (Vernier height gauge). In more detail, the distortion is quantified by the angle $\alpha(°)$ given by Equation (2):

$$\alpha = sin^{-1}\left[\frac{h_1 - h_2}{b}\right] \qquad (2)$$

where $\alpha$ = distorted angle, $h_1$ = total height at Vernier height, $h_2$ = total height of the workpiece from the surface, and *b* = length of the plate.

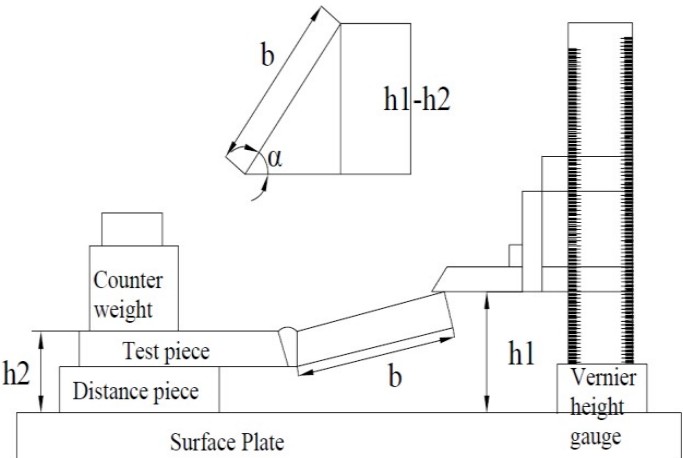

**Figure 3.** Schematic showing the welded joint disposition for the measurement of the distortion via the Vernier height gauge and definition of the distortion angle $\alpha(°)$.

During the joining process, the weld area experiences heating and cooling cycles so that differential thermal expansion and contraction of the weld metal and parent material cause welding residual stresses (RS). In this work, RS is measured in a transverse direction (perpendicular to the weld line) at various distances from the weld line on both the top and bottom surfaces of the weldment. Before measuring, the sample surface was cleaned with emery sheets of 20 mm × 20 mm. The Pulstec μ-X360n Portable X-ray Residual Stress analyzer, used for measurements, is shown in Figure 4a. The $\cos(\alpha)$ measuring method with a spot size of 2 mm is adopted; the camera image is displayed in Figure 4b. Sample measuring parameters are the diffraction angle (=150.876°), interplanar spacing ($d$ = 1.077 Å), X-ray wavelength (Cr) K-alpha (=2.29093 Å) and K-Beta (=2.08480 Å).

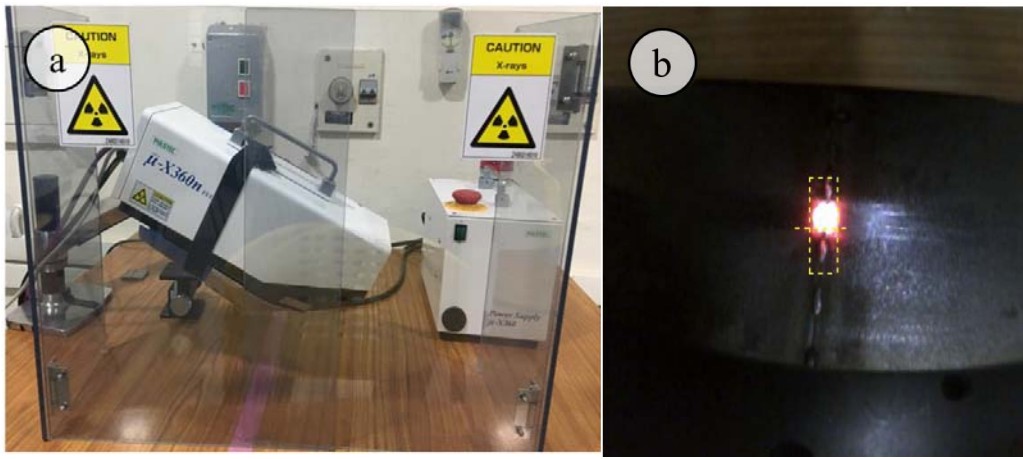

**Figure 4.** Residual stress measurement: (**a**) residual stress equipment, (**b**) camera image for sample location.

The Vickers microhardness test was performed on the weldments following the ASTM E-384-16 Standards. In particular, a 1 kg load was employed, while the indentation dwell time was about 8 s. Microhardness profiles were obtained for the weld bead cross-section near the root, middle, and cap bead region. In more detail, the profiles were obtained at a distance of 0.8, 2.5, and 4.2 mm from the bead root. Tensile tests were performed according to ASTM E8-15a Standards. Cutting of samples from the welded plates was carried out using the wire-cut electrical discharge machine (EDM).

## 3. Results and Discussion

### 3.1. Defects and Metallurgy

The weldments were found by visual inspection free from surface defects. NDT X-ray radiography tests detected defects such as macroporosity in the fusion zone (FZ) or incomplete penetration (Figure 5). ASME SEC IX-2017 standard was used for the quality testing of the weldments. It was observed that all weldments had straight white lines indicating acceptable weldments. Underfill and some porosities were identified in trial 1, while incomplete penetration was observed in trial 2 and trial 4; moreover, to better verify the weld's internal quality, macrographs of the weld bead cross-section were obtained from all trials and deeply exanimated.

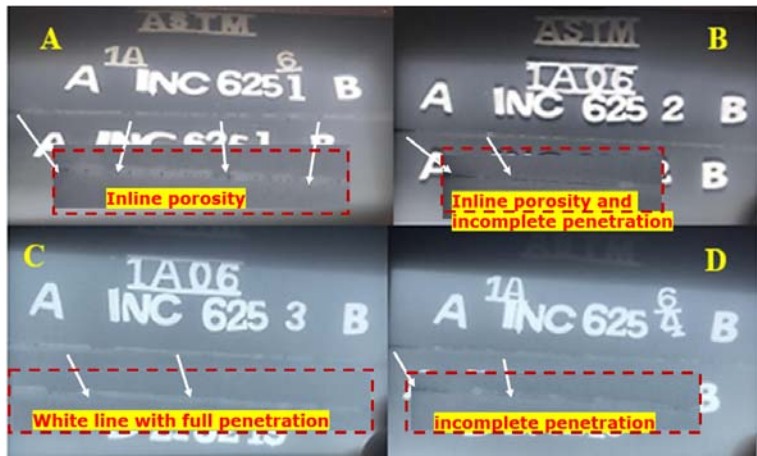

**Figure 5.** Weldments radiographic images (**A–D**).

Welding of high-strength alloys is always challenging in welding research aimed at achieving good metallurgical and mechanical properties by reducing the heat input. The observed macrostructure as a function of process parameters is shown in Figure 6. Weld pool dimensions were carefully measured. Full penetration at the highest heat inputs values of 180 and 198 J/mm was observed. In trial 1, in-line gas porosity was found (Figure 7), whereas trial 3 reached full penetration without detectable porosity. The weld pool width at the cap is wider and decreases at the root. Table 3 summarizes the main geometrical parameters of the FZ. It is observed that the weld pool's Y-shape is due to the laser beam diffraction promoted by the plasma formed at the top of the plate, called 'plume'. It is worth noting that the depth to width ratio is less than 2 (trials 1 and 3), which is much lower than that reached by arc welding processes (often greater than 2). A very narrow heat-affected zone characterizes all joints (Figures 7c and 8c), while the FZ microstructure shows the typical dendritic microstructure grown in the heat dissipation direction (Figures 7b and 8b). Moreover, the typical epitaxial grain growth is observed at the FZ/HAZ interface, as illustrated in Figures 7c and 8c.

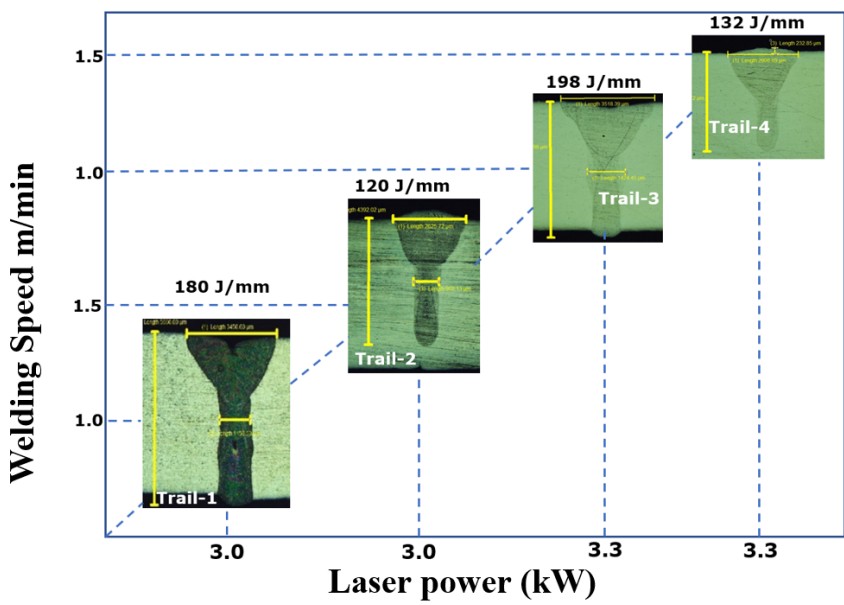

**Figure 6.** Macrographs of Inconel 625 laser beam welded joints as a function of process parameters (laser power and welding speed).

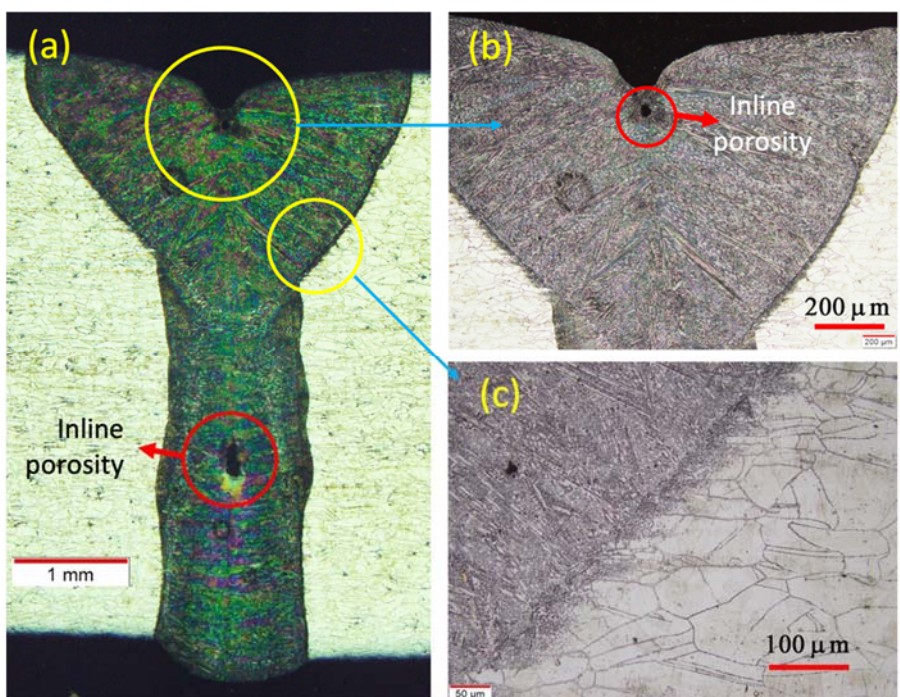

**Figure 7.** (**a**) Macrograph of Inconel 625 laser beam welded joint (weld 1) showing gas porosity on FZ and optical microscope images of the (**b**) weld cap and (**c**) HAZ.

**Table 3.** Weld pool geometry as a function of heat input.

| Trial No. | Heat Input [J/mm] | Weld Pool Width (Cap) [mm] | Weld Pool Height [mm] | Weld Pool Width (Root) [mm] | Penetration |
|---|---|---|---|---|---|
| 1 | 180 | 3.3 | 5.5 | 1.1 | Full |
| 2 | 120 | 2.6 | 4.3 | 1.2 | Partial |
| 3 | 198 | 3.5 | 5.4 | 1.4 | Full |
| 4 | 132 | 2.3 | 3.9 | 0.9 | Partial |

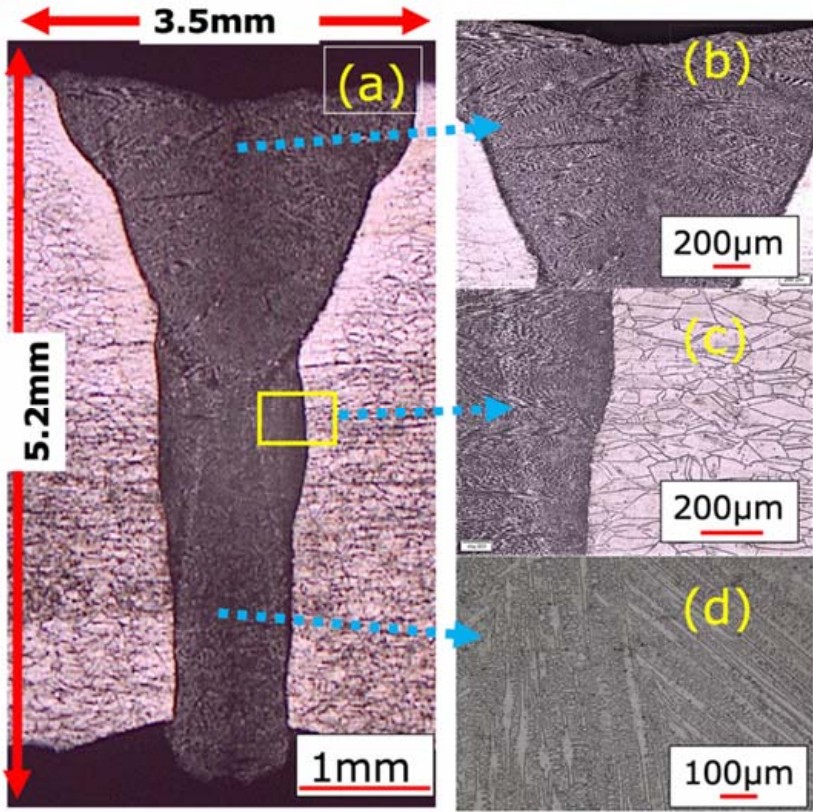

**Figure 8.** (**a**) Macrograph of Inconel 625 laser beam welded joint (weld 3) showing a defect-free weld with full penetration and optical microscope images of the (**b**) weld cap, (**c**) HAZ, and (**d**) weld root.

No solidification or liquation cracking at the FZ/HAZ interface occurred in the present Inconel 625 butt-welded joints. This is a consequence of the lower heat input of the $CO_2$ laser beam process compared to arc welding processes where, in contrast, such defects were detected, as reported by Harinadh [3], Ramkumar [26,30], Venkata Ramana [31] and Manikandan [32]. The microstructure is refined at the FZ/HAZ interface and became coarse, moving toward the weld centerline as a consequence of the competitive grain growth phenomenon, so that favorably oriented (FO) grains, whose preferred growth angle relative to the temperature gradient direction is small, grow preferentially relative to unfavorably oriented (UO) grains, which are characterized by a larger preferred growth angle. The XRD analysis as shown in Figures 9 and 10 revealed the precipitate of secondary phases in the fusion zone. The precipitates observed are $Cr_2Ni_3$, $Mn_3Si$, $Al_2FeSi$, $NbNi_3$ and $Cr_2Ni_3$, $Fe_7NI_3$, $Al_2FeSi$, $NbNi_3$ in weld 1 and weld 3, respectively. It is evident from the XRD analysis that Ni was enriched in the weld zone [12,24]. The major element in Inconel 625 is the weld bead, indicating intense peaks at 44.78°, 52.7°, and 76.35°. Cr and Ni are major constituents, and the precipitates of Cr and Ni were observed both in weld 1 and weld 3.

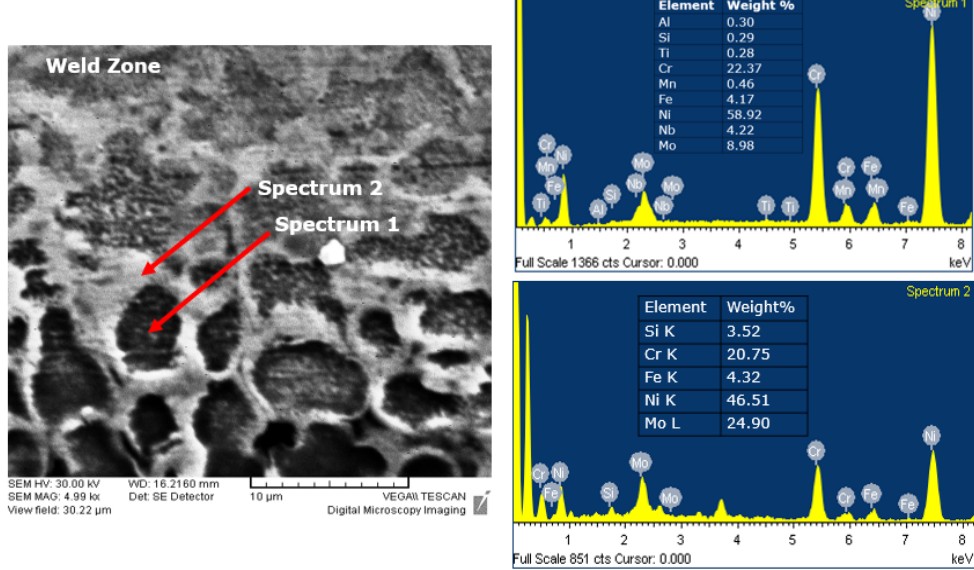

**Figure 9.** Back scattered SEM micrograph and EDS spectrum of the particles marked by arrows on sample 3.

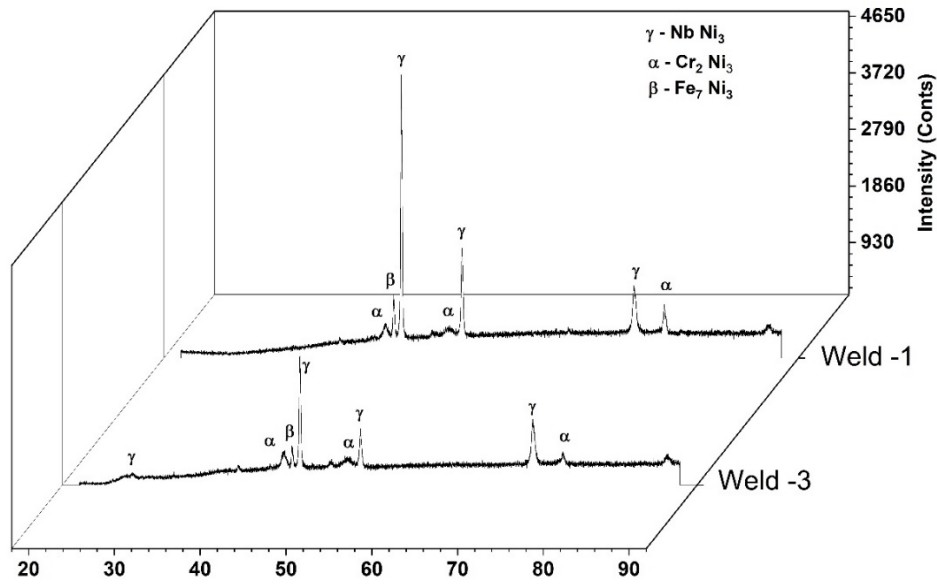

**Figure 10.** XRD analysis on FZ of $CO_2$ laser welded joints as a function of heat input (due to the use of Cr radiation, peaks of the Ni-based solid solution are not easily detectable).

### 3.2. Microhardness Profiles

Microhardness profiles covering the FZ, HAZ, and parent metal in cross-sections of welds 1 and 3 are shown in Figure 11. Firstly, we observed a slight increase in microhardness by approaching the fusion zone attributed to the finer microstructure in that zone compared to parent metal [17,25,26]. It was not possible to highlight the HAZ via microhardness profiles because of its very narrow size. In more detail, for weld 1, the microhardness values at the fusion zone in the cap, middle, and root zone are 255, 263, and 270 HV, respectively, whereas, in weld 3, the values are 255 HV at the cap and 270 HV at middle and root zone. Finally, the average microhardness value at the fusion zone was 262 HV.

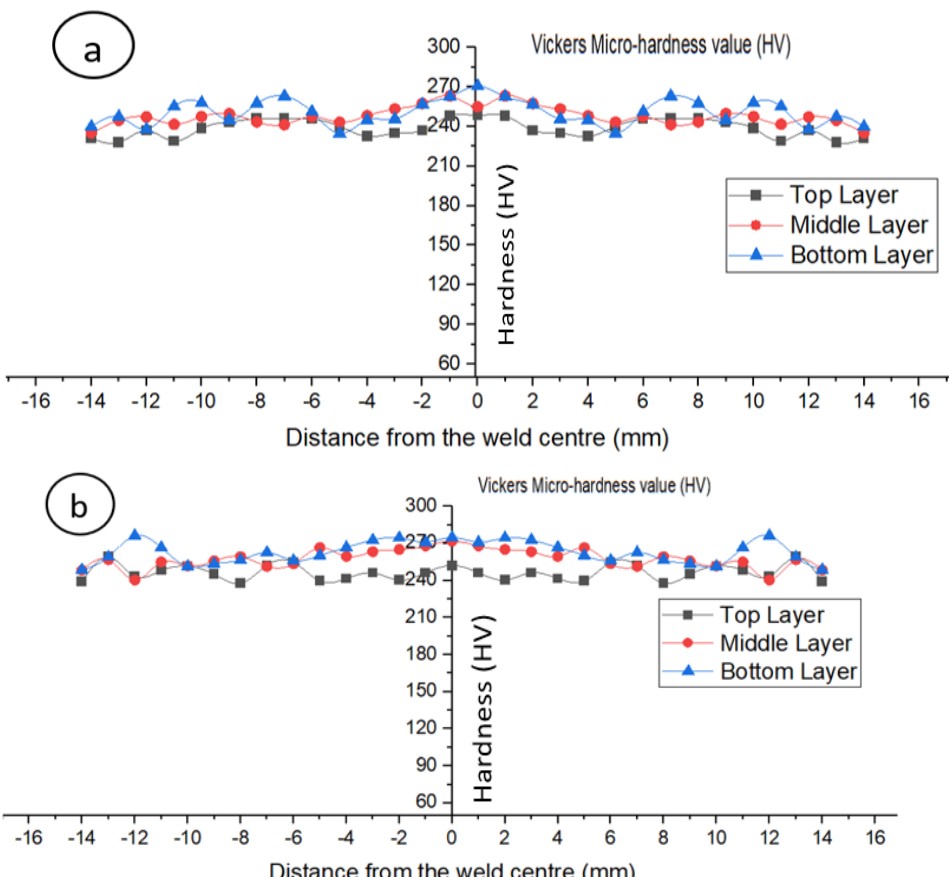

**Figure 11.** Microhardness profiles on joint cross-section; (**a**) trial 1, (**b**) trial 3.

### 3.3. Tensile Strength

Tensile tests were carried out at room temperature using specimens taken from weld 1 and weld 3. The ultimate tensile strength (UTS) is found to be 811 and 853 MPa for weld 1 and weld 3, respectively, while the yield stresses (YS) are 551 and 559 MPa, respectively (Figure 12a). Failures are located close to the weld line, as shown in Figure 12b. Details about crack initiation and propagation will be presented in a future work. Since the UTS and YS of the parent material are about 827 and 414 MPa, respectively [27], good mechanical properties were found, proving the excellent laser beam weldability of Inconel 625, provided that optimal process parameters are used. The obtained results are supported by Janicki's work [33], who found that a proper selection of laser welding parameters allows for obtaining the yield strength and ultimate tensile strength of the joints essentially equivalent to those of the base material.

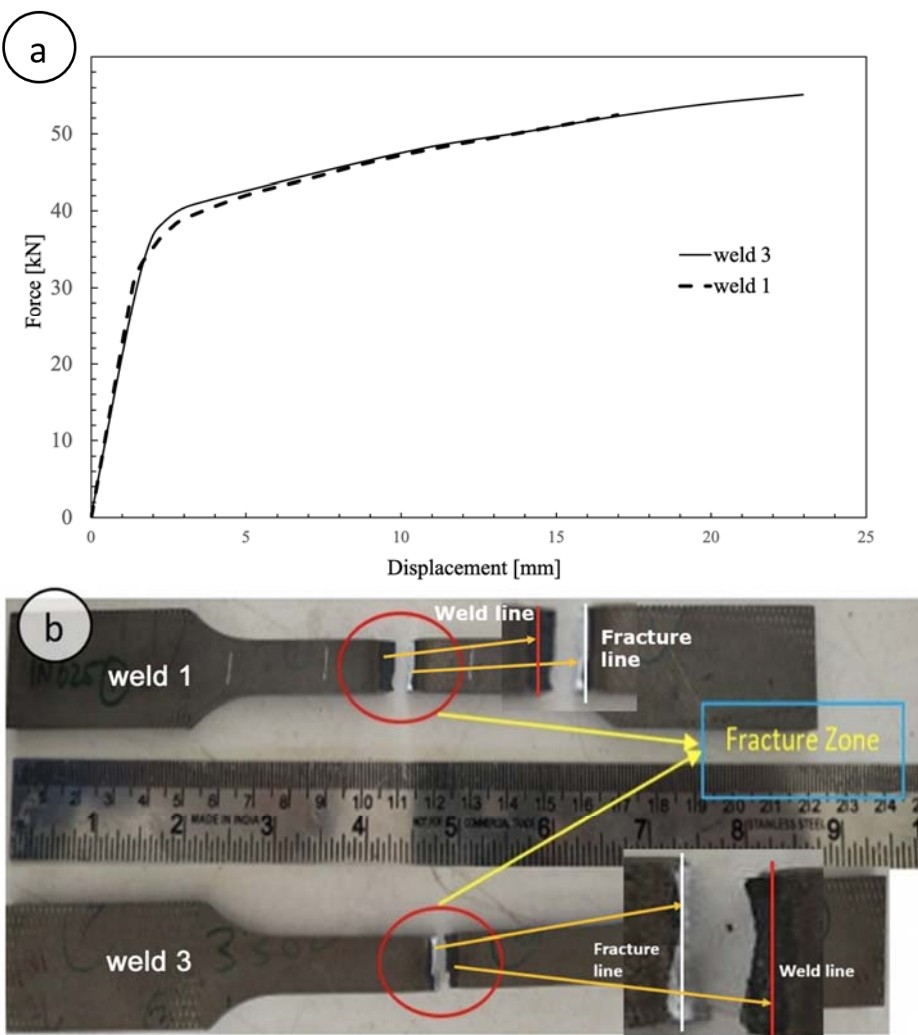

**Figure 12.** Tensile tests results: (**a**) UTS and YS, (**b**) failure location.

### 3.4. Distortion and Residual Stresses

The Vernier height gauge was used to measure the distortion in the laser weldments, defined as the difference between the heights $h_1$ and $h_2$, as schematized in Figure 3. Distortion was measured over the top surface after the weldment attained the ambient temperature. As expected, the higher the heat input, the higher the weld distortion. Weld 1, with a heat input of 180 J/mm, shows a distortion of about 3.57°, whereas weld 3, obtained with a heat input of 198 J/mm, is found to have a distortion of 4.45°. Therefore, an increase of about 1° is observed when moving from 180 to 198 J/mm heat input.

The residual stresses were measured on the top and bottom surfaces of the weldment using XRD techniques. The measurements were carried out in the transverse direction and were focused on the stress component perpendicular to the weld line, since this is of greater interest when considering the fatigue strength of welded joints [32,33]. Figure 13 shows the residual stress distribution measured on the top and bottom surfaces of the final weldments. Tensile residual stresses [34–40] were found throughout the plate. M-shaped stress distribution profiles are observed for weld 1 and weld 3 according to the literature [33]. The highest residual stress (158 MPa) was measured on joint 3 that, compared to weld 1, was characterized by higher heat input. Moreover, for each sample, the maximum residual stresses were measured on the bottom surface. While the maximum residual stresses were measured in the HAZ of joint 1 (116 MPa), in joint 3 the highest residual stress arose in FZ, showing a reverse M-shaped pattern. This result is quite surprising and suggests that residuals stresses are highly sensitive to welding process parameters

and boundary conditions. If the residual stresses on the top surface of joint 1 and joint 2 are compared, tensile (57 MPa) versus compressive (−41 MPa) stresses are observed in FZ [31,34].

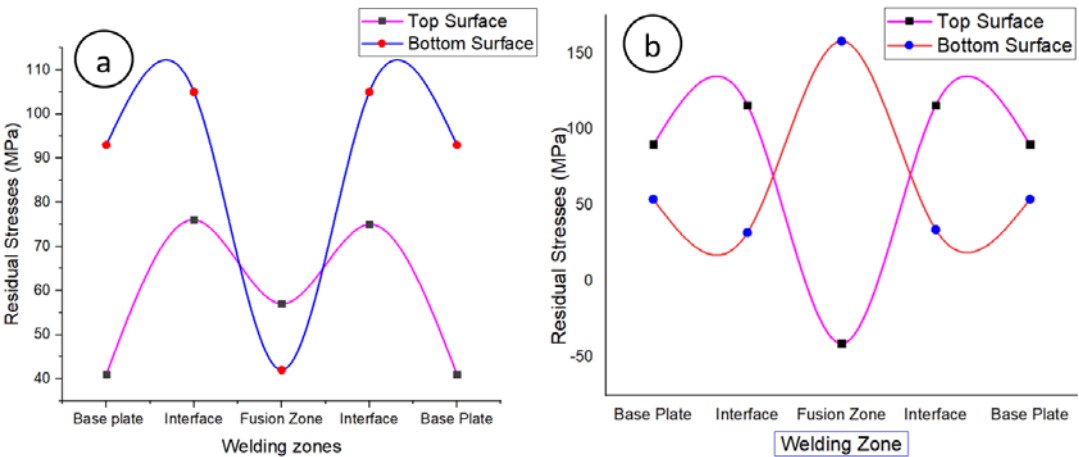

**Figure 13.** Residual stress distribution of the top and bottom surface of the final welds: (**a**) joint 1 and (**b**) joint 3.

## 4. Conclusions

Laser beam welding of Inconel 625 was investigated. Melt run trials were first carried out to optimize the process parameters. Almost defect-free specimens were then deeply analyzed via metallurgical investigation, residual stress and deformation measurements, and mechanical tests. The main outcomes can be summarized as follows

1. Full penetration of 5 mm thick Inconel 625 plates butt joint was obtained using a laser power of both 3 and 3.3 kW and a welding speed of 1.0 m/min. These values correspond to a heat input of 180 and 198 J/mm, respectively, which are much lower compared to those used in conventional fusion welding processes for the same material and geometry.
2. Metallurgical investigation showed a typical epitaxial growth in the fusion zone with eventually some porosity and an extremely narrow heat-affected zone.
3. The microhardness values in the weld bead of all specimens were found slightly greater than those measured on parent metal; this effect was attributed to the fusion zone finer microstructure compared to that of the parent metal.
4. Despite the presence of some porosity in the fusion zone, the tensile strength of the joints was comparable to that of the parent metal, indicating excellent laser weldability of such kind of alloy,
5. The higher the heat input, the higher the residual stress and distortion.
6. The results obtained in this paper will be used, in future work, for calibrating a laser beam welding process numerical model.

**Author Contributions:** Conceptualization, H.V., R.K.B. and E.G.; methodology, H.V., S.A. and R.K.B.; investigation, H.V., E.G. and A.C.U.M.R.; resources, H.V. and E.G.; writing—original draft preparation, H.V., A.C.U.M.R. and P.F.; writing—review and editing, H.V., P.F. and F.B. All authors have read and agreed to the published version of the manuscript.

**Funding:** This research received no external funding.

**Institutional Review Board Statement:** Not applicable.

**Informed Consent Statement:** Not applicable.

**Data Availability Statement:** Data is contained within the article.

**Acknowledgments:** The authors are gratefully acknowledged for the provision of equipment from the Science and Engineering Research Board, Department of Science and Technology, New Delhi,

under the Young Scientist Scheme (Ref. No. SB/FTP/ETA-0190/2014). The authors are also thankful to the management of the Vellore Institute of Technology (VIT), Vellore for enabling us to conduct Residual stress measurements using Pulstec Residual stress analyzer test in Advanced Material Processing and Testing Lab.

**Conflicts of Interest:** The authors declare no conflict of interest.

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
