# Peer review of "Mechanical and Metallurgical Properties of CO2 Laser Beam INCONEL 625 Welded Joints"

_applsci, doi:10.3390/app11157002_

Round 1

Reviewer 1 Report

Abstract - No clearly written work goal.
Research methodology and conclusions are written incorrectly.

spelling - for example line 105. Check out the full article.

Tables - units should be written in parentheses, some are incorrectly written, e.g. lit/min,  j / mm
This is just one of the few mistakes. Please check the article.

Figure 3 (graphics) and the caption under the drawing are to be improved.
The name INCONEL 625 is spelled differently throughout the article. Take 1 record and use it.
Figure 8 is too clipped from above.
Figure 10 - incorrectly presented diagram. no units, no scale.
Bad graphic notation and description under the picture.
Figure 11 - Incorrect axis record, it is in the wrong place.
What are "compressive (-41 MPa) stresses"? where does the minus come from?
No results for simulation of FEA, EBW. A reference in the literature is developed. 

Author Response

The authors express their sincere gratitude to the Editor/Reviewer for making valuable comments on the manuscript submitted. The suggestions and the mistakes pointed out by the editor are now addressed and the manuscript has been revised. 

Reviewer 2 Report

The paper deals with welding of Inconel625 produced by CO2 Laser beam. Microstructural and mechanical properties are investigated and reported. The whole welding process was optimized based on these results.

The paper is well structured and prepared. The theme of the paper has both scientific and technological interest and deserves to be published.

Some minor remarks

-The introduction section is too long. The paragraph referring to FEA can be ommitted since the authors do not deal with modeling in this paper.

- In fig. 5 in would be better in the author provide also images of higher magnification because it is difficult to distinguish the porosity in figure 5a.

- Figure 12: It is advised to present the stress-strain curves that can give better information than the bar chart.

- Figure 9: Is the SEM image a backscatted electon image? Please clarify

- The authors should better explain  the different residual stress distributions presented in Fig. 13. They should also refer to other scientific works to find out if this behavior is also observed by other researchers

Author Response

(The authors gave the same response as above.)

Reviewer 3 Report

Dear editor,

The article “Mechanical and Metallurgical Properties of CO2 Laser Beam 2 INCONEL 625 Welded Joints” could be published if a minor revision is made. Please find below my comments to the article.

Comments:

In line 137 “Figure 2 –“ needs to be deleted.

Line 208: Trial-1 contains not only pores, but also such a defect as weld underfill which has to be mentioned.

Figure 7(c) and Figure 8(c) have to be mirrored.

Figure 10 does not show any peaks of the matrix Ni-based solid solution. Presumably, it is caused by using of Cr radiation. If this is done on purpose, then it should be described in section 2.

In lines 280, 281 and 281 wrong units are used, replace them with “J/mm”

In section 3.4 there is a misprint “characterized”.

In section 5, paragraph 5 the word “and” should be deleted.

Author Response

(The authors gave the same response as above.)

Reviewer 4 Report

Please see attached

Author Response

(The authors gave the same response as above.)

Round 2

Reviewer 1 Report

Unclear entry on lines 135, 147;
line 190 - spelling; dot;
line 237 - Ni's - invalid spelling;
Figure 10 - XRD diffractogram still not resolved. The 3 high diffraction lines are not marked and not resolved.

Author Response

(The authors gave the same response as above.)

Reviewer 4 Report

Thankyou for making the corrections. It looks better now. But i have one fundamental issue about the paper which is still not well clarified.

Line 269-270: "It is worth noting that specimens taken from weld 1 failed at  269 parent metal, while those coming from weld 3 broke at HAZ ".

This is not evident in the figure 12b. Although, scale is not there in Figure 12b, Figure 8 indicate the weld length is 3.5 mm. It looks like failure occur in weld for both welds in Figure 12b. Metallography required to confirm this claim. The stress strain curve from Figure 12a also indicate failure may occur in similar location since it is not statistically shown that they are different. 

In addition, there are still number of typos. For example: figure 9 caption "back scatted" which should be "back scattered".

Author Response

(The authors gave the same response as above.)

Round 3

Reviewer 4 Report

I am not convinced with the modification of the updated draft. Metallography has to performed to justify the statement of line 275-276.

"It is worth noting that specimens taken from weld 1 failed at 
parent metal, very close to the HAZ, while those coming from weld 3 broke at HAZ (Figure  12b)."

Author Response

Comment reviewer 4, if metallographic analysis is not performed, the sentence "It is worth noting that specimens taken from weld 1 failed at parent metal, very close to the HAZ, while those coming from weld 3 broke at HAZ (Figure  12b)" should be revised to take into account that to confirm this, it will be necessary to apply this kind of metallographic analysis.

Response 

We agree with the reviewer comment therefore we decided to change that sentence because no metallographic analysis is available right now to confirm that finding.

The sentence was modified as follows:

Failures are located close to the weld line as shown in Fig. 12b. Details about crack initiation and propagation will be faced in the next work.

We appreciate a lot the effort spent by the reviewer aimed at improving our work.

Thank you.
